# Effects of *Cinnamomum zeylanicum* (Ceylon cinnamon) extract on lipid profile, glucose levels and its safety in adults: A randomized, double-blind, controlled trial

**Dimuthu Muthukuda**[1]**, Chamini Kanatiwela de Silva**[2]**, Saumiyah Ajanthan**[2]*****,
**Namal Wijesinghe**[2]**, Anuradha Dahanayaka**[2]**, Arunasalam Pathmeswaran**[3]

**1** Sri Jayawardenepura General Hospital, Nugegoda, Sri Lanka, **2** RemediumOne, Colombo, Sri Lanka,
**3** Faculty of Medicine, University of Kelaniya, Ragama, Sri Lanka

* saumiyahajanthan@remediumone.com

## Abstract

### Background

Cinnamon has been studied as a possible way to control blood glucose and serum cholesterol levels. However, there are no well-conducted randomized controlled trials that can accurately measure the lipid and glucose-lowering effects of *Cinnamomum zeylanicum* (*C. zeylanicum*) extract. This study primarily aimed to evaluate the effect of a standardized *C. zeylanicum* extract on serum low-density lipoprotein cholesterol (LDL-C) levels and secondarily on other lipid parameters (high-density lipoprotein cholesterol (HDL-C), total cholesterol and triglycerides), glucose levels, anthropometric measures, blood pressure, and safety outcomes in individuals with an LDL level between 100-190mg/dL.

### Materials and methods

This was a randomized, double-blinded, placebo-controlled clinical trial. Participants were allocated to either *C. zeylanicum* extract or placebo group (1:1 allocation ratio). They were advised to take two capsules per day (1000 mg/day, a dose based on prior clinical studies suggesting potential efficacy and safety). Reduction in LDL-C at 12 weeks (from the baseline value) was compared between the two groups using ANCOVA. A complete-case analysis was adhered to in analyzing the outcome data.

### Results

The mean age (SD) of the 150 participants was 50.4 (10.52) years, and 66% were females. Among the 127 participants assessed at 12 weeks, those in the *C. zeylanicum* extract arm had a lower LDL-C value than the placebo arm but the difference was not significant (the baseline adjusted mean difference was 6.05mg/dL; 95% CI: -2.43 to 14.52; p = 0.161). However, participants in the *C. zeylanicum* extract group showed significantly greater reductions in fasting blood sugar (FBS) levels (the baseline adjusted mean difference was 8.59mg/dL; 95% CI: 0.59 to 16.59; p = 0.036). There was a significant interaction effect between the

**Data Availability Statement:** All relevant data are within the paper and its Supporting Information files.

**Funding:** Financial support for this research was provided by SDS Spices Pvt Ltd. The funders had no role in study design, data collection and analysis, decision to publish, or preparation of the manuscript.

**Competing interests:** I have read the journal's policy and the authors of this manuscript have the following competing interests: Financial support for this research was provided by SDS Spices Pvt Ltd. This funding covered the costs associated with staff at research sites, including the reimbursement of authors Dimuthu Muthukuda and Arunasalam Pathmeswaran for the time devoted to this study. The authors declare that this funding did not influence the study design, data collection, analysis, interpretation, or the decision to publish these findings. We confirm that our adherence to PLOS ONE policies on sharing data and materials is not altered by the competing interests statement. Further, this does not alter our adherence to PLOS ONE policies on sharing data and materials.

supplement and participants' glycemic status, with individuals with type 2 diabetes mellitus (T2DM) who received *C. zeylanicum* extract experiencing a notable reduction in FBS levels (standardized coefficient: -63, 95% CI: -102 to -25; p = 0.002).

## Conclusions

*C. zeylanicum* extract did not have significantly reduce LDL-C but demonstrated a significant FBS-lowering effect, particularly in individuals with T2DM, with a favorable safety profile.

## Trials registration

The trial was registered with the Sri Lanka Clinical Trials Registry: SLCTR/2021/011.

## Introduction

Complementary and Alternative Medicines (CAMs) include a wide variety of medical and therapeutic practices that fall outside the scope of conventional medicine. Research has increasingly focused on the use of CAMs among patients with chronic conditions such as diabetes, cancer, and cardiovascular disease [1–4]. The high utilization of CAMs in these populations is influenced by multiple factors, including cultural beliefs, limited access to conventional healthcare, and the often-prohibitive costs associated with conventional medical treatments [5–7].

Cinnamon, a longstanding traditional remedy in various cultures, has been used to treat a wide array of conditions, including headaches, toothaches, colds and gastrointestinal issues [8,9]. A recent review of 14 clinical trials on humans reported that cinnamon can contribute as an antioxidant and an anti-inflammatory agent [10]. However, scientific findings on cinnamon's efficacy have been mixed, and there is a need for further investigation to clarify its role and mechanisms of action.

Dyslipidemia, characterized by abnormal lipid levels in the blood, is a major modifiable risk factor for cardiovascular disease. Elevated low-density lipoprotein cholesterol (LDL-C) levels, in particular, have been associated with increased mortality and morbidity [11]. Similarly, type 2 diabetes mellitus (T2DM) is a significant risk factor for cardiovascular disease, with "diabetic dyslipidemia" serving as a crucial link between diabetes and cardiovascular complications [12]. Effective management of glycemic levels and lipid profiles is essential for reducing the risk of cardiovascular events and improving overall health [13,14].

The South Asian population exhibits a higher prevalence of cardiometabolic risk factors compared to other ethnic groups, even at lower body mass index (BMI) levels. his phenomenon is linked to an increased risk of premature coronary heart disease [15]. Furthermore, their unique lipoprotein profile contributes significantly to their higher risk of atherosclerotic cardiovascular disease [16].

Coumarins are naturally occurring plant compounds with strong anticoagulant properties that can potentially exert toxic effects on the liver [17]. Most clinical trials on cinnamon have utilized *Cinnamomum cassia*, which is characterized by its high coumarin content. In contrast, research involving *Cinnamomum zeylanicum* (*C. zeylanicum*), known for its lower coumarin content, remains limited [18,19]. To our knowledge, only one clinical trial has been conducted

using a full spectrum extract of *C. zeylanicum* (meaning both polar and non-polar compounds), but it was an acute study [20].

To date, there are no well-conducted randomized controlled trials that quantify the lipid lowering and glucose lowering effects of *C. zeylanicum* extract in humans, as well as to determine whether these effects have any other health implications. Although certain in vitro, in vivo, and preliminary clinical studies suggest that *C. zeylanicum* may lower serum lipid levels and confer potential cardiovascular benefits, these findings are inconclusive due to the lack of rigorously designed trials. This randomized, double-blinded, placebo-controlled trial was designed to address these gaps by evaluating the potential effects of *C. zeylanicum* extract on serum LDL-C in individuals with an LDL-C level between 100–190 mg/dL.

## Materials and methods

### Trial design

A randomized, double-blinded, placebo-controlled clinical trial was conducted at the Sri Jayawardenepura General Hospital and the Kandawala Medical Centre in Sri Lanka from 3 May 2021 to 29 March 2022. The last patient visit took place on 31 May 2022 and the trial sites were closed out on 29 August 2022. The study was approved by the Ethics Review Committee of Faculty of Medicine, University of Kelaniya, Sri Lanka and registered at the Sri Lanka Clinical Trials Registry (SLCTR/2021/011), which is a primary registry linked to the Registry Network of the International Clinical Trials Registry Platform of the WHO (WHO-ICTRP). The clinical trial was conducted in compliance with the Declaration of Helsinki and the Good Clinical Practice (GCP) guidelines.

### Inclusion and exclusion criteria

The participants were identified from a cohort of patients with an LDL-C level between 100–190 mg/dL (100 mg/dL < LDL-C < 190 mg/dL) and were aged 18 to 70 years. Participants were excluded if they, 1) had a history of allergy to cinnamon; 2) were already on cinnamon or any other nutritional/ herbal/ ayurvedic supplements; 3) were lactating, pregnant or unwilling to use an effective form of birth control for women of childbearing years; 4) had any form of malignancy at screening or in the past; 5) had blood dyscrasias; 6) eGFR <60ml/min/1.73m$^2$; 7) was diagnosed with alcoholic liver disease (ALD), decompensated cirrhosis or abnormal baseline liver function tests; 8) had cardiac, liver, renal or respiratory failure; 9) had atherosclerotic cardiovascular disease (ASCVD) or any other major critical illnesses; 10) were on statins or any other lipid lowering drug; 11) have triglycerides level >300 mg/dL or 12) had a history of epilepsy and/or were on anti-epileptic drugs.

Initially, the approved protocol excluded participants with an estimated glomerular filtration rate (eGFR) <30 ml/min/1.73m$^2$, rather than <60 ml/min/1.73m$^2$. Additionally, individuals with a history of epilepsy or currently using anti-epileptic drugs were not excluded. However, to prioritize patient safety, these exclusion criteria were revised. It's noteworthy that prior to these adjustments, no participants with an eGFR <30 ml/min/1.73m$^2$ or with a history of epilepsy or using anti-epileptic drugs were enrolled in the study.

### Randomization

The subjects were allocated to either cinnamon intervention or placebo group with an allocation ratio of 1:1. The randomization sequence was generated using the 'blockrand' package in R [21], ensuring that the allocation process was both random and balanced. Block randomization was employed with block sizes of 4 and 6, which helps to maintain balance in the number

of participants allocated to each group. The varying block sizes were used to further prevent predictability of allocation, thus enhancing the integrity of the randomization process.

Central randomization was implemented in this study for the two sites to ensure unbiased and systematic allocation of participants to different treatment groups. When a participant was enrolled at either of the two study sites, the recruiting staff contacted an independent central methods center. The central methods center, which was not involved in participant recruitment or treatment, accessed the randomization list and provided the next sequentially numbered container that corresponded to the treatment assignment. This process ensured that the allocation of participants was done systematically and unbiasedly, without influence from the local recruiting staff.

## Blinding

Allocation concealment was achieved by using pre-packed, sequentially numbered containers. These containers were identical in appearance and were prepared by an independent party, ensuring that neither the participants nor the study personnel (including those at the central methods center) knew the contents of each container (cinnamon or placebo).

## Investigational medicinal product

The investigational medicinal product (IMP) was a capsule containing a standardized cinnamon bark extract. The Ceylon cinnamon extract tested in this study is manufactured by SDS Spices (commercially available under the CeyCinnX trademark). Extraction was carried out with aqueous + ethanol solvent. The concentrated output was passed through a vacuum dryer at specific temperatures to obtain a powder (which is standardized to a minimum of 30% polyphenols constituting of largely proanthocyanidin type A epicatechin polymers (PAC A).

The test product contained cinnamaldehyde ($0.99 \pm 0.00$ (%,w/w)), eugenol ($1.85 \pm 0.02$ (%, w/w)) and cinnamyl acetate ($34.72 \pm 0.63$ (%,w/w)) as major compounds, as quantified by using high-performance liquid chromatography method. Also, the cinnamon extract contained 15298 mg of PAC A per 100g (measured via BL-DMAC method). The extract, derived from Ceylon cinnamon, contained low levels of coumarin, measured at 0.01%.

The placebo capsule was composed of pharmaceutical grade wheat flour (tasteless, odorless white color fine powder). It was identical in shape, size, weight and texture to the *C. zeylanicum* capsule. To mask the distinctive aroma of cinnamon, cinnamon quills were placed in the packets containing both the placebo and the cinnamon capsules.

Each participant included in the treatment arm was advised to take two capsules per day (1000 mg/day); one capsule (500 mg) before breakfast and one capsule before dinner. The dose was selected based on prior preclinical and clinical studies suggesting potential efficacy and safety [20,22]. The investigator prescribed the same dose to be taken after meals (after breakfast and after dinner) if a participant showed any sign of intolerance due to taking the capsule before meals. The capsule was administered orally.

As this was a double-blinded study, the random allocation information was not available to the study participants or the investigators.

## Sample size calculation

The sample size was calculated using the formula given below, based on the assumption that the IMP would reduce LDL-C by 10mg/dL, which was the minimum difference expected to be of clinical significance. Assuming a standard deviation of 40mg/dL for LDL-C [23] and a correlation of 0.9 between the baseline and 12 weeks measurements, a sample size of 64 per group

would give 90% power at a significance level of 0.05. Before accounting for the attrition rate, the sample size needed was 128 participants (64 per group). Expecting a 15% dropout rate, the final sample size required was 150 (75 per arm) [24].

$$n \approx \left( \frac{4\sigma^2(1-\rho^2)\left(z_{1-\frac{\alpha+z1-\beta}{2}}\right)}{\Delta} \right)^2$$

Where:

- $\Delta$ is the expected reduction in LDL-C (10 mg/dL),

- $\sigma$ is the standard deviation of LDL-C (40 mg/dL),

- $\rho$ is the correlation coefficient between baseline and 12-week measurements (0.9),

- $z_{1-\frac{\alpha}{2}}$ is the critical value for a significance level of 0.05,

- $z_{1-\beta}$ is the critical value for 90% power.

## Outcome measures

Data collection was carried out by trained research assistants, using a standardized case record form, at baseline and during each 4-weekly follow-up visit. A brief history was taken from each patient, which included symptoms related to any disease, and subsequently, a physical examination was carried out.

The outcomes of the study were assessed at baseline and at the end of the 12-week study period. The primary outcome measure was the change in LDL-C levels from baseline to the final follow-up, 12 weeks post-randomization.

Secondary outcome measures included changes in other lipid profile parameters (total cholesterol, triglycerides and HDL-C), changes in FBS and Hemoglobin A1c (HbA1c), changes in anthropometric parameters (height, weight, waist and hip circumferences), and changes in systolic blood pressure (SBP) and diastolic blood pressure (DBP).

The study also evaluated the potential effects of regular administration of *C. zeylanicum* extracts on liver (aspartate aminotransferase (AST), alanine transaminase (ALT)) and kidney function (estimated glomerular filtration rate (eGFR), serum creatinine), as well as the occurrence of self-reported adverse events in participants.

## Statistical analysis

Means and standard deviations (SD) were calculated for all outcome measures at baseline and for changes from baseline to 12 weeks. Mean differences between the *C. zeylanicum* and placebo groups were evaluated using t-tests, both for unadjusted comparisons and for comparisons adjusted for baseline values. Confidence intervals (CI) and p-values were calculated to evaluate the statistical significance and precision of the mean differences and reductions. Additionally, effect size indices, such as Cohen's d, were calculated to quantify the magnitude of the differences between groups.

The reduction in LDL-C levels and other clinical parameters over 12 weeks, from baseline, was compared between the two groups—*C. zeylanicum* and placebo—using regression analysis. The dependent variable was the change in the clinical parameters of interest from baseline to 12 weeks, while the independent variable was the treatment group (*C. zeylanicum* vs. placebo). The treatment group was coded as a binary variable, with the *C. zeylanicum* group coded as 1 and the placebo group as 0.

Little's MCAR test was performed to assess the missing data pattern. It indicated that the data were missing completely at random (MCAR), allowing us to proceed with a complete-case analysis for the outcome data. Analysis of covariance (ANCOVA) was used to assess changes in LDL-C levels and other clinical parameters at 12 weeks, with the treatment group as the main factor and age as a covariate. Prior to conducting the ANCOVA, we assessed the assumptions, including the homogeneity of regression slopes, the normality of residuals, and the homogeneity of variances. All assumptions were met, ensuring the validity of the analysis.

**Exploratory analysis.** An interaction analysis was conducted to examine potential interaction effects between the treatment (*C. zeylanicum* vs. placebo) and participants' baseline glycemic status (normoglycemia, prediabetes, and T2DM). A two-way ANOVA was employed to determine whether the treatment effect on FBS levels at 12 weeks varied across glycemic status subgroups.

Data analysis was done using version 4.0 of the R statistical software [25].

## Results

### Baseline characteristics

The total number of subjects recruited for the study was 150, out of which 127 completed the 12 weeks follow up (**Fig 1**).

The mean (SD) age of the participants was 50.4 (10.5) years, and 66% of participants were females. Mean (SD) body mass index (BMI) was 26.3 (4.51) kg/m², while waist circumference and hip circumference were 93.3 (10.40) cm and 100.2 (10.41) cm, respectively. Baseline lipid profiles revealed mean (SD) total cholesterol, LDL-C, HDL-C, and triglyceride were 205.3

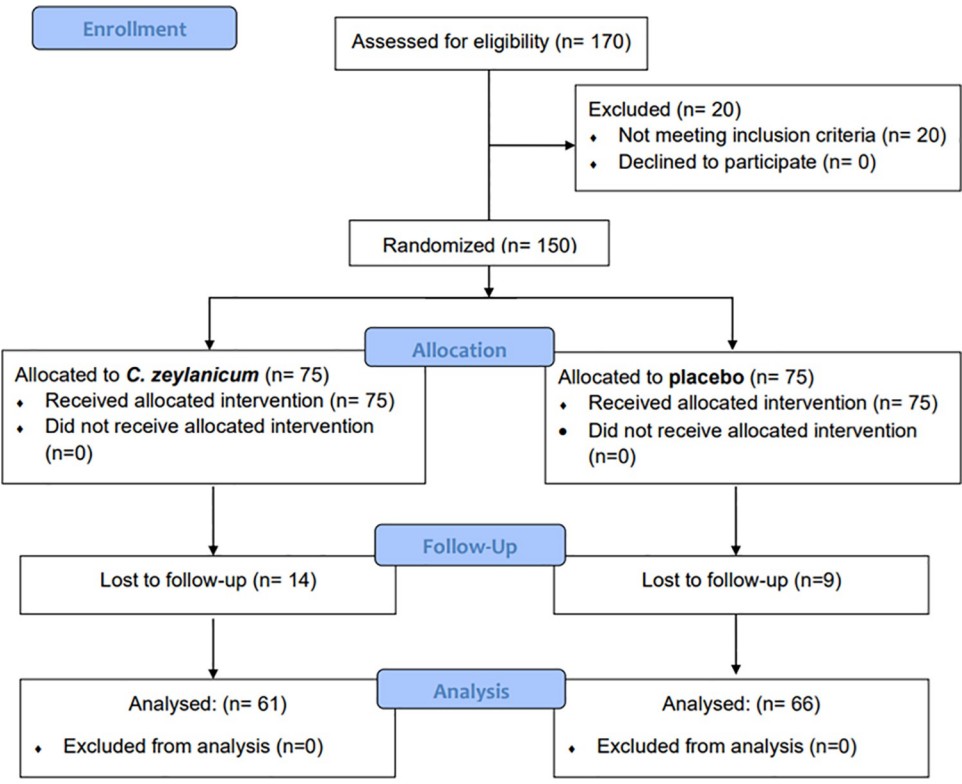

**Fig 1. Subject disposition throughout the trial.**

**Table 1. Baseline characteristics of study participants.**

| Characteristic | Placebo<br>N = 75[a] | C. zeylanicum<br>N = 75[a] |
|---|:---:|:---:|
| Age, years | 52.04 (10.85) | 48.73 (9.99) |
| Sex | | |
| Female | 50.0 (67.0) | 49 (65.0) |
| Male | 25.0 (33.0) | 26 (35.0) |
| **Comorbidities** | | |
| Hypertension | 10.0 (13.0) | 6.0 (8.0) |
| Glycaemic status[b] | | |
| Normoglycemia | 39.0 (52.0) | 42.0 (56.0) |
| Prediabetes | 21.0 (28.0) | 21.0 (28.0) |
| T2DM | 15.0 (20.0) | 12.0 (16.0) |
| **Self-reported use** | | |
| Tobacco consumption | | |
| Current Smoker | 1.0 (1.3) | 1.0 (1.3) |
| Ex-smoker | 3.0 (4.0) | 0.0 (0.0) |
| Never smoked | 71.0 (95.0) | 74.0 (99.0) |
| Alcohol Consumption, Yes | 5.0 (6.7) | 2.0 (2.7) |
| Caffeine Consumption, Yes | 54.0 (72.0) | 51.0 (68.0) |
| **Clinical parameters** | | |
| LDL-C, mg/dL | 134.36 (20.82) | 132.58 (23.13) |
| HDL-C, mg/dL | 44.51 (7.14) | 43.86 (6.31) |
| Triglycerides, mg/dL | 133.75 (41.16) | 144.01 (52.74) |
| Total Cholesterol, mg/dL | 205.53 (25.74) | 205.11 (28.80) |
| FBS, mg/dL | 108.55 (38.10) | 114.05 (62.19) |
| HbA1c, % | 5.90 (0.99) | 5.98 (1.40) |
| AST, U/L | 25.41 (10.69) | 27.13 (13.16) |
| ALT, U/L | 38.34 (21.01) | 37.66 (22.34) |
| Serum creatinine, mg/dl | 0.84 (0.21) | 0.82 (0.18) |
| eGFR, mL/min/1.73m$^2$ | 90.21 (17.19) | 93.79 (16.82) [c] |
| **Anthropometrics** | | |
| BMI, kg/m$^2$ | 25.94 (3.87) | 26.72 (5.07) |
| Waist circumference, cm | 93.10 (10.52) | 93.58 (10.35) |
| Hip circumference, cm | 99.19 (10.03) | 101.16 (10.75) |
| Mid-arm circumference, cm | 28.76 (3.45) | 29.78 (4.30) |

[a] Mean (SD); n (%).

[b] Glycaemic level classification: *Normoglycemia* (HbA1c < 5.7%); *Prediabetes* (HbA1c: 5.7 to 6.5%); *T2DM* (HbA1c > 6.5%).

[c] eGFR was not recorded for one participant.

(27.22) mg/dl, 133.4 (21.95) mg/dl, 44.2 (6.72) mg/dl, and 138.9 (47.43) mg/dl respectively. The mean FBS was 111.3 (51.48) mg/dl. Additionally, at baseline, the mean (SD) SBP, DBP, AST, ALT and eGFR were 130.1 (19.05), 83.5 (10.24), 26.3 (11.98) U/L, 38.0 (21.62) U/L and 92.0 (17.04) mL/min/1.73m$^2$, respectively.

The baseline demographic and anthropometric characteristics of the participants randomized to each group are given in **Table 1**.

## Primary and secondary outcomes

Table 2 shows the results of the outcomes at baseline and after 12 weeks. Of the 127 participants with a 12 week assessment, those in the *C. zeylanicum* arm had a lower LDL-C value than the placebo arm but the difference was not significant (the baseline adjusted mean difference was 6.05 mg/dL; 95% CI: -2.43 to 14.52; p = 0.161; Table 2).

Participants in the *C. zeylanicum* extract group experienced significantly greater reductions in FBS levels compared to the placebo group, with a baseline-adjusted mean difference of 8.59 mg/dL (95% CI: 0.59 to 16.59; p = 0.036; Table 2). No significant differences were observed between the groups for HDL-C, triglycerides, total cholesterol, HbA1c, AST, ALT, eGFR, serum creatinine and BMI (Table 2).

Table 3 presents the results of univariate and multiple linear regression analyses examining the effects of *C. zeylanicum* extract, compared to placebo, on clinical outcomes. The primary outcome, LDL-C, decreased by an average of 4.6 mg/dL in the *C. zeylanicum* extract group compared to the placebo group. After adjusting for age, this decrease was slightly reduced to 3.88 mg/dL, remaining statistically non-significant (p = 0.395).

For FBS, a significant decrease was observed in both the unadjusted and adjusted models. The unadjusted model showed a decrease of 15.92 mg/dL in the *C. zeylanicum* extract group (p = 0.048), while the adjusted model showed a slightly larger decrease of 16.59 mg/dL (p = 0.042).

Other secondary outcomes, including HDL-C, triglycerides, total cholesterol, and measures of liver and kidney function, showed minimal differences between the unadjusted and adjusted models, with all results remaining statistically non-significant.

Most effect sizes were negligible, with a few indicating small differences between the *C. zeylanicum* and placebo groups for mean reductions. The only exception was FBS, where a small but significant effect was observed (Cohen's d = -0.3549; 95% CI: −0.7092, −0.0007; Table 3), indicating that *C. zeylanicum* may have a beneficial effect on lowering FBS levels.

## Adverse events

Over the 12-week period, the percentages of participants with any adverse events and serious adverse events were similar between the *C. zeylanicum* extract group and the placebo group (Table 4). Gastritis was the most prevalent adverse event in both study groups. In response, the investigator recommended administering the same dosage post-meals (following breakfast and dinner) for participants displaying signs of intolerance, such as gastritis, potentially caused by capsule ingestion before meals. The adverse events did not lead to a permanent dose reduction or discontinuation of IMP.

## Exploratory analysis

The mean change in FBS levels from baseline to 12 weeks was analyzed across different glycemic status groups (normoglycemia, prediabetes, and T2DM) for participants receiving either *C. zeylanicum* extract or a placebo.

For participants with T2DM, those treated with *C. zeylanicum* extract (N = 12) had a substantial mean FBS decrease of -78.6 mg/dL (SD = 108.52), whereas the placebo group (N = 15) showed a mean change of -11.2 mg/dL (SD = 55.47) (Table 5 and Fig 2).

The interaction between *C. zeylanicum* extract and glycemic status of the participant was significant. The standardised coefficient for the interaction term of *C. zeylanicum* extract * Prediabetes was -5.5 (95% CI: -38 to 27, p = 0.740; Table 6), indicating no significant effect. However, for *C. zeylanicum* extract × T2DM, the standardised coefficient was -63 (95% CI: -102 to -25, p = 0.002; Table 6), demonstrating a significant reduction in FBS levels among

**Table 2. The mean difference in outcomes after 12 weeks of treatment with either a placebo or *C. zeylanicum* extract.**

| Parameter | Placebo N = 61[a] | *C. zeylanicum* N = 66[a] | Mean difference at 12 weeks | | Effect Size (95% CI) [b,e] |
|---|---|---|---|---|---|
| | | | Placebo minus *C. zeylanicum* (95% CI, p-value) [b,c] | Adjusted for Baseline (95% CI, p-value) [b,d] | |
| LDL-C, mg/dL | 138.46 (29.09) | 129.82 (26.52) | 8.64 (-1.16, 18.45) p = 0.084 | 6.05 (-2.43, 14.52) p = 0.161 | -0.3110 (−0.6646, 0.0426) |
| HDL-C, mg/dL | 45.15 (5.13) | 44.06 (4.90) | 1.09 (-0.68, 2.85) p = 0.225 | 0.40 (-0.94, 1.74) p = 0.558 | -0.2169 (−0.5694, 0.1357) |
| Triglycerides, mg/dL | 130.79 (39.25) | 146.71 (55.61) | -15.93 (-32.74, 0.89) p = 0.063 | -12.01 (-27.40, 3.38) p = 0.125 | 0.3287 (−0.0252, 0.6825) |
| Total Cholesterol, mg/dL | 210.05 (32.88) | 201.80 (32.84) | 8.25 (-3.31, 19.80) p = 0.160 | 5.54 (-3.67, 14.76) p = 0.236 | -0.2510 (−0.6038, 0.1019) |
| FBS, mg/dL | 105.72 (30.93) | 99.85 (22.21) | 5.87 (-3.67, 15.41) p = 0.225 | 8.59 (0.59, 16.59) **p = 0.036** | -0.2195 (−0.5721, 0.1330) |
| Hemoglobin A1c, % | 5.82 (1.03) | 5.83 (1.05) | -0.01 (-0.38, 0.35) p = 0.955 | 0.05 (-0.11, 0.21) p = 0.509 | 0.0101 (−0.3414, 0.3616) |
| AST, U/L | 27.48 (13.20) | 27.64 (13.16) | -0.16 (-4.79, 4.47) p = 0.945 | 1.35 (-2.12, 4.83) p = 0.442 | 0.0122 (−0.3393, 0.3637) |
| ALT, U/L | 40.36 (25.64) | 36.20 (19.51) | 4.16 (-3.90, 12.22) p = 0.308 | 3.42 (-1.73, 8.56) p = 0.191 | -0.1837 (−0.5360, 0.1685) |
| eGFR, mL/min/1.73m$^2$ | 91.47 (17.78) | 94.29 (16.85) | -2.83 (-8.92, 3.27) p = 0.360 | -0.17 (-5.07, 4.72) p = 0.944 | 0.1635 (−0.1886, 0.5156) |
| Serum creatinine, mg/dl | 0.85 (0.20) | 0.82 (0.15) | 0.03 (0.04, 0.09) p = 0.387 | 0.019 -0.027, 0.065 p = 0.407 | -0.1559 (-0.5079, 0.1962) |
| Waist circumference, cm | 91.44 (10.72) | 92.58 (9.94) | 1.142 (-4.78, 2.50) p = 0.536 | -0.64 (-1.51, 0.22) p = 0.144 | 0.1107 (−0.2411, 0.4625) |
| Hip circumference, cm | 97.65 (10.02) | 100.07 (10.33) | 2.421 (-6.00, 1.15) p = 0.183 | -0.38 (-1.42, 0.67) p = 0.476 | 0.2378 (−0.1150, 0.5905) |
| Mid arm circumference, cm | 28.21 (3.73) | 29.42 (4.37) | 1.211 (-2.63, 0.21) p = 0.095 | -0.63 (-1.28, 0.02) p = 0.057 | 0.2973 (−0.0561, 0.6508) |
| Weight, Kg | 65.87 (12.72) | 66.85 (11.21) | -0.973 (-5.20, 3.25) p = 0.649 | -0.086 (-1.09, 0.91) p = 0.865 | 0.0814 (0.2703, 0.4330) |
| BMI, kg/m$^2$ | 25.93 (4.05) | 26.37 (5.19) | -0.44 (-2.07, 1.19) p = 0.594 | -0.02 (-0.39, 0.35) p = 0.908 | 0.0940 (−0.2577, 0.4457) |

[a] Mean (SD).

[b] CI = Confidence Interval.

[c] Welch Two Sample t-test.

[d] ANCOVA.

[e] Cohen's d.

**Table 3. Univariate and multiple linear regression analyses of the effects of *C. zeylanicum* extract (compared to placebo), on changes in clinical outcomes from baseline to 12 weeks.**

| Characteristic | Mean reduction at 12 weeks (95% CI) [a] | | | Unadjusted model | | Adjusted model (adjusted for age) | | |
|---|---|---|---|---|---|---|---|---|
| | Placebo N = 61 | *C. zeylanicum* N = 66 | Effect Size [b] | Standardized coefficients (95% CI) [a] | p-value | Standardized coefficients (95% CI) [a] | p-value | Adjusted $R^2$ |
| **Primary Outcome** | | | | | | | | |
| LDL-C, mg/dL | 2.56 (-3.8, 9.0) | -2.01 (-8.3, 4.2) | -0.1811 (−0.5333, 0.1711) | -4.57 (-13.44, 4.30) | 0.310 | -3.88 (-12.85, 5.10) | 0.395 | 0.0003 |
| **Secondary Outcomes** | | | | | | | | |
| **Lipid Profile Parameters** | | | | | | | | |
| HDL-C, mg/dL | -0.34 (-2.0, 1.3) | 0.04 (-0.94, 1.0) | 0.0713 (−0.2803, 0.4229) | 0.38 (-1.48, 2.24) | 0.689 | 0.49 (-1.39, 2.37) | 0.608 | -0.0100 |
| Triglycerides, mg/dL | -3.91 (-15, 7.4) | 2.88 (-11, 17) | 0.1306 (−0.2213, 0.4825) | 6.79 (-11.49, 25.08) | 0.464 | 7.82 (-10.73, 26.36) | 0.406 | -0.0077 |
| Total Cholesterol, mg/dL | 2.03 (-4.5, 8.6) | -2.47 (-9.5, 4.6) | -0.1657 (−0.5178, 0.1864) | -4.50 (-14.05, 5.05) | 0.353 | -3.00 (-12.54, 6.55) | 0.535 | 0.0234 |
| **Glycemic levels** | | | | | | | | |
| FBS, mg/dL | 0.39 (-7.6, 8.4) | -15.53 (-29, -2.1) | **-0.3549 (−0.7092, −0.0007)** | -15.92 (-31.69, -0.15) | **0.048** | -16.59 (-32.6, -0.59) | **0.042** | 0.0177 |
| HbA1c, % | -0.10 (-0.20, 0.01) | -0.17 (-0.33, -0.01) | -0.1361 (−0.4880, 0.2159) | -0.07 (-0.27, 0.12) | 0.445 | -0.08 (-0.28, 0.11) | 0.407 | -0.0094 |
| **Liver and kidney function** | | | | | | | | |
| ALT, U/L | 2.05 (-0.43, 4.5) | 0.13 (-2.6, 2.8) | -0.1859 (−0.5381, 0.1664) | -3.23 (-8.58, 2.12) | 0.234 | -3.58 (-9.00, 1.84) | 0.193 | 0.0010 |
| AST, U/L | 1.70 (-2.1, 5.5) | -1.53 (-5.3, 2.3) | -0.2123 (0.5648, 0.1402) | -1.92 (-5.56, 1.71) | 0.297 | -2.05 (-5.74, 1.64) | 0.274 | -0.0057 |
| eGFR, mL/min/1.73m[2] | 1.11 (3.3, 5.5) | -0.14 (-3.4, 3.1) | -0.0827 (−0.4357, 0.2703) | -1.25 (-6.61, 4.10) | 0.644 | -1.63 (-7.04, 3.79) | 0.553 | -0.0071 |
| Serum creatinine, mg/dl | 0.01 (-0.03, 0.06) | 0.00 (-0.03, 0.03) | -0.0934 (−0.4451, 0.2583) | -0.01 (-0.07, 0.04) | 0.600 | -0.01 (-0.07, 0.04) | 0.659 | -0.0119 |
| **Anthropometrics** | | | | | | | | |
| Waist circumference, cm | -0.94 (-1.6, -0.26) | -0.32 (-0.89, 0.26) | 0.2497 (-0.1031, 0.6026) | 0.62 (-0.25, 1.50) | 0.162 | 0.72 (-0.16, 1.60) | 0.109 | 0.0158 |
| Hip circumference, cm | -0.51 (-1.3, 0.32) | -0.32 (-1.0, 0.40) | 0.0627 (−0.2889, 0.4143) | 0.19 (-0.89, 1.28) | 0.725 | 0.19 (-0.91, 1.29) | 0.735 | -0.0151 |
| Mid arm circumference, cm | -0.65 (-1.1, -0.24) | -0.07 (-0.59, 0.44) | 0.3075 (-0.0461, 0.6611) | 0.58 (-0.08, 1.24) | 0.086 | 0.58 (-0.09, 1.25) | 0.091 | 0.0077 |
| Weight, Kg | -0.71 (-1.3, -0.13) | -0.68 (-1.5, 0.15) | 0.0124 (−0.339, 0.3639) | 0.04 (-0.99, 1.06) | 0.945 | 0.10 (-0.93, 1.14) | 0.846 | -0.0106 |
| BMI, kg/m[2] | -0.26 (-0.48, -0.04) | -0.25 (-0.54, 0.05) | 0.0118 (−0.3397, 0.3633) | 0.01 (-0.36, 0.38) | 0.947 | 0.04 (-0.34, 0.41) | 0.842 | -0.0098 |
| **Vitals** | | | | | | | | |
| Systolic blood pressure, mmHg | 4.25 (1.2, 7.3) | 1.52 (-1.4, 4.5) | -0.0017 (−0.3532, 0.3498) | -2.74 (-6.91, 1.43) | 0.196 | -1.87 (-5.99, 2.25) | 0.371 | 0.0542 |
| Diastolic blood pressure, mmHg | 1.23 (-1.6, 4.0) | 1.21 (-1.1, 3.5) | -0.2307 (−0.5834, 0.1220) | -0.02 (-3.55, 3.52) | 0.992 | 0.03 (-3.56, 3.62) | 0.987 | -0.0159 |

[a] CI = Confidence Interval.

[b] Cohen's d.

**Table 4. Adverse events in the overall population.**

| Event | Placebo (N = 75) | *C. zeylanicum* (N = 75) |
|---|---|---|
| Any adverse event, n (%) | 6 (8.0) | 5 (7.7) |
| Most frequent adverse events, n (%) | | |
| Gastritis | 4 (5.3) | 4 (5.3) |

participants with T2DM compared to the placebo group. This interaction accounted for 6.8% of the variation in the FBS levels, increasing the model's R-squared from 17.3% (without interaction) to 24.1% (with interaction).

## Discussion

We conducted a multicenter, randomized, double-blinded, placebo-controlled trial, to evaluate the potential lipid-lowering and other effects of *C. zeylanicum* extract in 150 individuals from Sri Lanka over 12 weeks. While our study demonstrated a reduction in LDL-C levels in the treatment group compared to the placebo group, this reduction was not statistically significant.

Our findings align with a systematic review and meta-analysis of 13 randomized controlled trials involving 750 participants, which found no significant effect of cinnamon supplementation on LDL-C or HDL-C levels [26]. In contrast, a phase I clinical trial conducted by Ranasinghe et al., involving 28 healthy adults, demonstrated that *C. zeylanicum* extract (administered in escalating doses of 85 mg, 250 mg, and 500 mg capsules over 12 weeks) significantly reduced total cholesterol ($p < 0.05$) and LDL-C ($p < 0.001$) [22]. Similarly, a recent study involving 40 patients with stage 1 hypertension found significant improvements in total cholesterol and LDL-C levels with a 1500 mg/day dose of *C. zeylanicum* extract over 90 days (total cholesterol decrease: 10.1±10 mg/dl, p = 0.001 and LDL-C decrease: 19.2 ± 24 mg/dl, p = 0.001) [27]. However, these studies included participants with broader inclusion criteria, not specifically those with elevated LDL-C as in our study. Additionally, they had smaller sample sizes compared to ours.

Our study reported a significant reduction in fasting blood sugar (FBS) values in participants receiving *C. zeylanicum* extract (Table 3; unadjusted model: p = 0.048; age-adjusted model: p = 0.042). In contrast, no significant difference was observed in average HbA1c levels (Table 3; unadjusted model: p = 0.445; age-adjusted model: p = 0.407). The significant reduction in FBS, even in the unadjusted model, suggests that *C. zeylanicum* extract may have a notable effect on short-term glucose control.

Previous research has demonstrated that cinnamon, rich in polyphenolic components, reduces oxidative stress and improves pre-prandial glucose levels when consumed at 500 mg/

**Table 5. Mean change in FBS levels from baseline to 12 weeks across different glycemic status groups (Normoglycemia, prediabetes, and T2DM) for participants receiving either *C. zeylanicum* extract or a placebo.**

| Glycemic status[1] | *C. zeylanicum* | | Placebo | |
|---|---|---|---|---|
| | N | Mean FBS change (SD) | N | Mean FBS change (SD) |
| Normoglycemia | 42 | -1.81 (15.15) | 39 | 2.4 (14.07) |
| Prediabetes | 21 | -5.19 (29.44) | 21 | 4.53 (30.29) |
| T2DM | 12 | -78.6 (108.52) | 15 | -11.2 (55.47) |

[1] Glycaemic level classification: *Normoglycemic* (HbA1c < 5.7%); *Prediabetes* (HbA1c: 5.7 to 6.5%); *T2DM* (HbA1c > 6.5%).

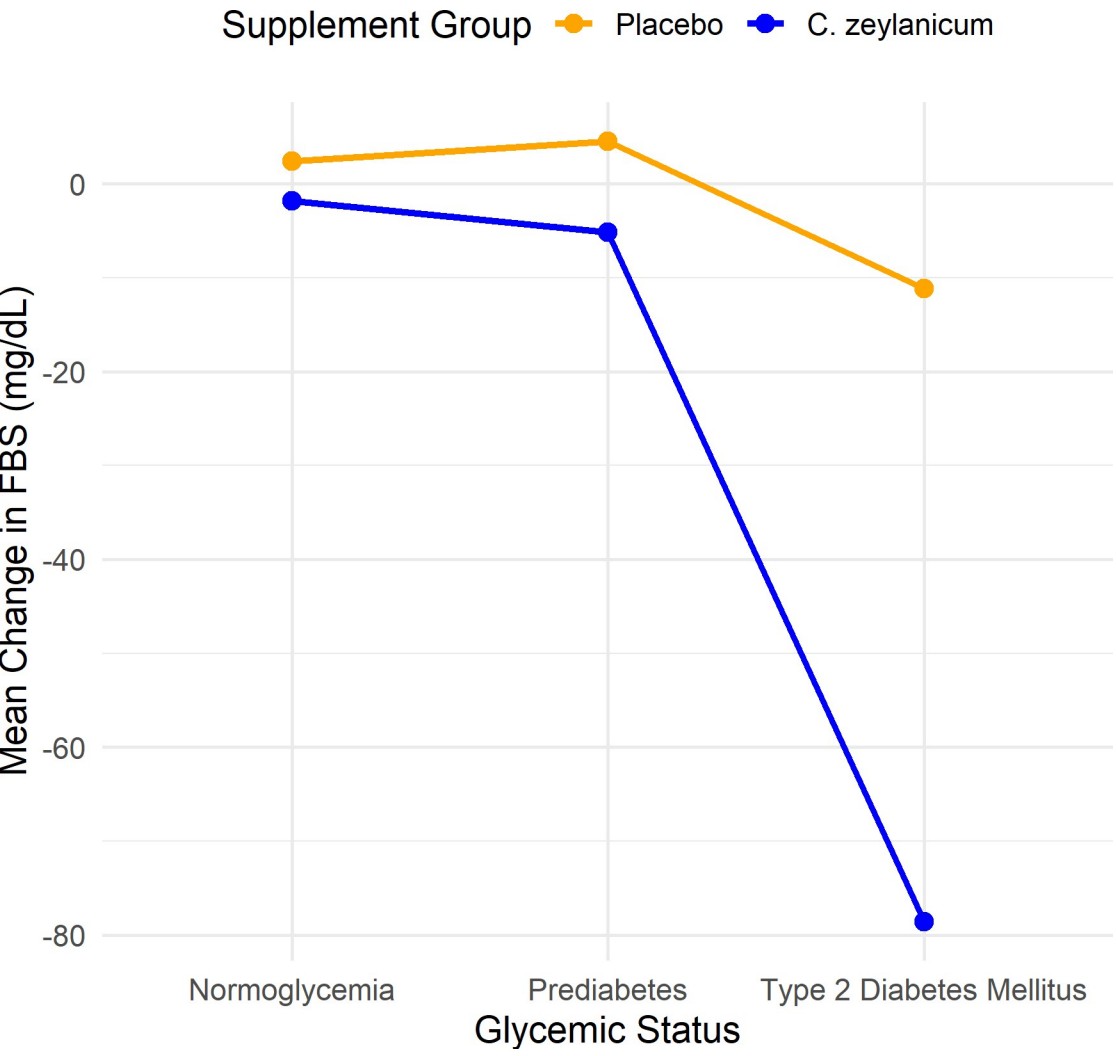

**Fig 2. Effect of glycemic status and supplement group on change in FBS.**

day for 12 weeks [28]. A trial carried out in 2018 aimed at investigating the effects of various amounts of cinnamon on factors including pre-prandial blood glucose and HbA1c levels in 41 healthy volunteers over 40 days found an average 5.92% reduction (p = 0.035) in pre-prandial blood sugar levels among those consuming 6g of cinnamon daily [29]. This trial also reported no statistically significant difference in average HbA1c levels, consistent with our findings.

A notable finding from our study is the significant interaction between the supplement (*C. zeylanicum* vs Placebo) and participant's glycemic status on FBS level (p = 0.002, Table 6). As illustrated in Fig 2, individuals diagnosed with T2DM experienced the greatest decrease in FBS when given *C. zeylanicum* extract. Thus, the effect of the supplement on the FBS value depends on the glycemic status of the participant. A study conducted in China with participants with T2DM (n = 66) reported a statistically significant reduction in FBS levels in the low-dose group, which consumed 2g of cinnamon per day. FBS levels decreased from 9.00 to 7.99 mmol/L, with an average reduction of 1.01 mmol/L (p = 0.002) [30]. In contrast, our study observed a reduction in FBS levels in participants with T2DM who consumed a lower dose of cinnamon (1g per day). Similarly, another study involving 60 participants with T2DM found

**Table 6. Summary of multiple linear regression analysis for interactions between supplement (*C. zeylanicum* vs. placebo) and glycemic status explaining changes in FBS levels from baseline to 12 weeks.**

| Characteristic | Standardised coefficient | 95% CI[1] | p-value |
|---|---|---|---|
| **Supplement** | | | |
| Placebo | — | — | |
| *C. zeylanicum* | -4.2 | -24, 15 | 0.671 |
| **Glycemic status** | | | |
| Normoglycemia | — | — | |
| Prediabetes | 2.1 | -21, 26 | 0.858 |
| T2DM | -14 | -41, 14 | 0.327 |
| **Supplement * Glycemic status** | | | |
| *C. zeylanicum* * Prediabetes | -5.5 | -38, 27 | 0.740 |
| *C. zeylanicum* * T2DM | -63 | -102, -25 | **0.002** |

[1]*CI = Confidence Interval.*

*$R^2$ = 0.241.*

that even 1 g of cinnamon significantly reduced FBS levels, from 11.6 ± 1.7 mmol/L to 8.7 ± 1.6 mmol/L over 40 days [31]. However, the cinnamon used in that study was *Cinnamomum cassia*, which contains higher amounts of coumarin compared to *C. zeylanicum* extract [19]. This distinction is important, as *Cinnamomum cassia* poses a greater risk of potential toxicity, making it a less safe option for regular consumption.

In vitro and in vivo models suggest that cinnamon may exert antihyperglycemic effects through insulin-mimetic actions and modulation of glucose metabolism pathways [32–34]. Research has indicated that T2DM is improved by cinnamon extract due to its capability of translocating GLUT4 through the AMPK signaling pathway [35]. Cinnamon extract has been shown to increase the amounts of insulin receptors (IR), insulin receptor substrates (IRS), and GLUT4 receptors, which collectively enhance the cellular uptake of glucose [36]. A study specifically highlighted that *C. zeylanicum* extracts significantly boost GLUT4 translocation and its production in the plasma membrane of adipose tissue [37].

Additionally, the proanthocyanidins (PACs) in *C. zeylanicum*, comprising at least 15% of the extract used in our trial, are emerging as key components in modulating glucose homeostasis. Higher PAC intake has been associated with reduced diabetes risk, though their exact mechanism remains unclear [38,39]. However, the underlying mechanism of action is still not understood, due to their complex chemical structure [39]. A study conducted by Jiao et al., found that cinnamon water extract (CWE) inhibited the amyloid formation of human islet amyloid polypeptide (hIAPP) in a dose-dependent manner, and identified PACs as the major anti-amyloidogenic compounds of CWE. Thus, the study results indicated a potential pharmacological usage of PACs as an anti-diabetic drug candidate [40].

Our study found that *C. zeylanicum* extract is safe for consumption, as it did not cause any significant changes in liver function (AST and ALT levels) or kidney function (eGFR), (Table 2) and supports the Phase 1 trial results of a study carried out in 30 healthy volunteers [22].

We did not observe any significant reductions in most anthropometric parameters (weight, BMI, waist circumference and hip circumference) during and after the 12 weeks of *C. zeylanicum* supplementation, similar to previous pre-clinical and clinical studies [22,37].

Future research should explore comparisons between *C. zeylanicum* and *Cinnamomum cassia*, particularly in terms of coumarin content and health implications. Additionally, comparing different extraction methods (e.g., water-based vs. ethanol extracts) could provide insights

into the optimal preparation for therapeutic use. Additionally, this is compounded by the fact that there are studies that have not identified the cinnamon species that is being utilized. In a 2018 article, Oketch-Rabah et al., discussed the issue of correct nomenclature of cinnamon, and how it presents a challenge to the applicability of clinical data. They go on to further add that due to differences in quality and composition of various *Cinnamomum* species, safety and efficacy data are not generalizable [41]. Our study, however, does not present such issues, as we have clearly stated the source as well as the method in which the capsules were made. Hence, if our intervention was compared to an intervention containing the same amount of *Cinnamomum cassia*, we would be able to investigate the safety benefits of Ceylon cinnamon in comparison to Chinese cinnamon, which is the next common alternative in current use.

While our study did not achieve the anticipated LDL-C reduction, it contributes valuable data on the blood glucose-lowering effects of *C. zeylanicum* extract. Given the higher prevalence of T2DM in the Asian Indian population, our findings underscore the potential role of Ceylon cinnamon in managing blood glucose levels.

The major strengths of this study include its relatively large sample size compared to previous research on *C. zeylanicum* extract, as well as its broad inclusion criteria and good statistical power to identify moderate intervention effects on clinically relevant outcomes. To our knowledge, this is the largest trial to date assessing the effects of *C. zeylanicum* extract on LDL-C. Limitations include missing data, which primarily resulted from participant loss to follow-up and thus the unavailability of data for certain outcomes or measurements. This may impact statistical power, suggesting the need for further research to confirm these results and explore additional aspects of cinnamon's effects.

## Conclusion

In conclusion, this trial did not find *C. zeylanicum* supplementation to have any significant effect on LDL-C. But it significantly reduced FBS levels. Our results did not identify any significant adverse effects or toxicity of *C. zeylanicum* extract. Further studies with larger samples and longer durations are likely to provide more definitive evidence on the effects of *C. zeylanicum*.

## Supporting information

**S1 File. Dataset.**
(XLSX)

**S2 File. Protocol.**
(PDF)

**S1 Text. CONSORT checklist.**
(DOC)

## Acknowledgments

The authors thank: the trial participants who agreed to take part in the trial; Buddhi Ashan Ratnawardana and Vindya Vishwabhashini Weerasinghe of SDS spices (Pvt) Ltd for ensuring the timely production, packaging and labelling of the capsules and placebos and delivery of the same; Buddhipraba Dananji Kariyawasam, Gayan Wijeratne and Dilini Fernando (RemediumOne), Attidiya Widanalage Manoja Nilanthi Karunarathne and Samanmali Dhammadinna Dissanayake (Sri Jayawardenepura General Hospital) for data collection; Thanushanthan Jeevaraja, Shehan Gnanapragasam and Chanaka Fernando (RemediumOne) for project start-up

activities; Kavindu Fernando, Enalee Ranasinghe, Chrishmi Rodrigo and Shalomi Weerawardana (RemediumOne) for trial monitoring.

## Author Contributions

**Conceptualization:** Dimuthu Muthukuda, Chamini Kanatiwela de Silva, Arunasalam Pathmeswaran.

**Data curation:** Namal Wijesinghe, Anuradha Dahanayaka.

**Formal analysis:** Saumiyah Ajanthan, Arunasalam Pathmeswaran.

**Methodology:** Dimuthu Muthukuda, Chamini Kanatiwela de Silva, Arunasalam Pathmeswaran.

**Project administration:** Anuradha Dahanayaka.

**Writing – original draft:** Saumiyah Ajanthan.

**Writing – review & editing:** Dimuthu Muthukuda, Chamini Kanatiwela de Silva, Namal Wijesinghe, Anuradha Dahanayaka, Arunasalam Pathmeswaran.

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
