## [Decision Letter · Decision Letter 0]

15 Jul 2024

PONE-D-24-00185Effects of Cinnamomum zeylanicum (Ceylon cinnamon) extract on lipid profile, glucose levels and its safety in adults: A randomized, double-blind, controlled trialPLOS ONE

Dear Dr. Ajanthan,

Thank you for submitting your manuscript to PLOS ONE. After careful consideration, we feel that it has merit but does not fully meet PLOS ONE’s publication criteria as it currently stands. Therefore, we invite you to submit a revised version of the manuscript that addresses the points raised during the review process.

We look forward to receiving your revised manuscript.

Kind regards,

Jennifer Tucker, PhD

Staff Editor

PLOS ONE

Journal Requirements:

   "Financial support for this research was provided by SDS Spices Pvt Ltd. "

   "I have read the journal's policy and the authors of this manuscript have the following competing interests:

Financial support for this research was provided by SDS Spices Pvt Ltd. This funding covered the costs associated with staff at research sites, including the reimbursement of authors Dimuthu Muthukuda and Arunasalam Pathmeswaran for the time devoted to this study. The authors declare that this funding did not influence the study design, data collection, analysis, interpretation, or the decision to publish these findings."

5. In the online submission form, you indicated that "The data that support the findings of this study are available from the corresponding author, S.A, upon reasonable request."

7. Please ensure that you refer to Figure 2 in your text as, if accepted, production will need this reference to link the reader to the figure.

8. Please include a separate caption for each figure in your manuscript.

Additional Editor Comments:

Please follow the reviewers recommendations for revision, including providing additional contextualization in the introduction and further methodological details. 

Comments from PLOS Editorial Office: We note that one or more reviewers has recommended that you cite specific previously published works. As always, we recommend that you please review and evaluate the requested works to determine whether they are relevant and should be cited. It is not a requirement to cite these works. We appreciate your attention to this request.

Reviewers' comments:

Reviewer's Responses to Questions

**Comments to the Author**

1. Is the manuscript technically sound, and do the data support the conclusions?

Reviewer #1: Partly

Reviewer #2: Yes

2. Has the statistical analysis been performed appropriately and rigorously? 

Reviewer #1: No

Reviewer #2: Yes

3. Have the authors made all data underlying the findings in their manuscript fully available?

Reviewer #1: Yes

Reviewer #2: Yes

4. Is the manuscript presented in an intelligible fashion and written in standard English?

Reviewer #1: Yes

Reviewer #2: Yes

5. Review Comments to the Author

Reviewer #1: Comments

For the sample size calculation, whether a formula or sample size calculator/software was used to derive the sample size is to be stated.

The information on the effect size, how the correlation coefficient of 0.9 between baseline and six-month measurements was incorporated into the sample size calculation, and the sample size before the attrition rate 15% is to be stated.

Line 173-174, although complete case analysis is easy to implement, it can contribute to biased results if the missing data are not missing completely at random (MCAR). Perhaps a statement on handling missing data could be added by exploring the pattern of missing data and the possible imputation method, model-based approaches etc if required.

Line 172-174, the ANCOVA analysis is unclear. More details are to be provided. The dependent variable, independent variable(s), and covariate variable are to be stated. Whether the dependent variable refers to LDL-C at 3 months or changes from baseline to 3 months is to be clearly stated. From Table 2 presentation, it indicates the dependent variable LDL-C is at 3 months.

A statement on the fulfillment of ANCOVA assumptions is to be provided.

Line 176-177, the treatment and potential subgroups are to be mentioned and the name of statistical test that will be employed is to be mentioned.

Line 178, the actual citation for the statistical software R is to be provided.

Table 1 Female, typo 6.07%.

Table 1, for the glycemic status variable, the subcategories are not to be directly below the same alignment with the glycemic status available. Since symbol % has been indicated in the first column of those variables, all individual symbol % could be omitted.

Table 2, the table requires cosmetic changes and to follow the journal requirement. Mean, SD, 95% CI, statistical test is to be denoted in the table footnote. For Hb available, AST, eGFR, ACR, BMI for p >0.9, the actual p-value is to be provided. The decimal point for the p-value is to be standardized in the table and text. One decimal point is to be avoided. This applies to others where applicable. The word mean fall is to be avoided and replaced with another word.

Line 212, P=0.036 to be replaced as p=0.036

Line 215-220, the adjusted findings are to be “discussed” as well apart from the unadjusted findings.

Effect size indices could be presented.

Table 3, how the independent variable was coded for the multivariate linear regression is to be stated. For the adjusted analysis, model fit is to stated.

Table 3, Beta is to be labeled as standardized coefficients and denoted in the table footnote.

Table 3 anthropometrics and vitals (for p >0.9), the actual p-value is to be presented. Similarly, the decimal point for the p-value, beta coefficient and 95%CI is to be standardized. One decimal point for the p-value is to be avoided. This applies to others where applicable.

Line 221, multivariate linear regression is to be stated in the statistical analyses section. The use of multivariate linear regression is to be clearly stated, e.g. whether the focus is on estimating treatment effects at follow-up etc and whether adjusting for baseline values is required.

Line 221-222, the word ‘at 3 months’ is to be stated.

Line 239, (-63), (95% CI to be presented as (-63, 95%C….)

Line 246, the timepoint baseline -3 months is to be stated.

Line 254, the timepoint or changes are to be stated.

Table 5 is to be cited in the text.

Table 6, the decimal point for the p-value is to be standardized. One decimal point is to be avoided.

References did not adhere to the format specified by the journal.

Reviewer #2: Dear Authors

I read your manuscript carefully; your work is acceptable with minor revise. Please apply the following comments.

Article entitled “Effects of Cinnamomum zeylanicum (Ceylon cinnamon) extract on lipid profile, glucose levels and its safety in adults: A randomized, double-blind, controlled trial” has been written in a good way; It is an interesting topic. Authors only address following minor revisions:

Abstract:

-Your Keywords should written based on MeSH term.

Introduction:

In the introduction, you should first discuss the role of complementary therapies (natural compounds) products in human health and chronic diseases prevention and treatment, also its role in improving glycemic indices and lipid profile levels in adults, then discuss the probable role of cinnamon in glycemic and lipid profile indices. You can also use the following references in your article for strengthen the introduction and discussion sections. These references are helpful.

1) Ranasinghe, P., Galappaththy, P., Constantine, G. R., et al. (2017). Cinnamomum zeylanicum (Ceylon cinnamon) as a potential pharmaceutical agent for type-2 diabetes mellitus: study protocol for a randomized controlled trial. Trials, 18, 1-8.

2) Gheflati et al. The effects of cinnamon supplementation on adipokines and appetite-regulating hormones: A systematic review of randomized clinical trials.

3) The effect of conjugated linoleic acids on inflammation, oxidative stress, body composition and physical performance: a comprehensive review of putative molecular mechanisms (Nutrition & Metabolism).

4) Does propolis have any effect on rheumatoid arthritis? A review study (Food Science & Nutrition).

5) Nattagh‐Eshtivani et al. The role of Pycnogenol in the control of inflammation and oxidative stress in chronic diseases: Molecular aspects.

6) Effects of Momordica charantia L on blood pressure: a systematic review and meta-analysis of randomized clinical trials (International Journal of Food Properties).

7) Bahari et al. The effect of Royal jelly on liver enzymes and glycemic indices: A systematic review and meta-analysis of randomized clinical trials.

8) Barghchi et al. The effects of Chlorella vulgaris on cardiovascular risk factors: A comprehensive review on putative molecular mechanisms.

Methods:

-Please more explain about randomization and blinding.

-Please mention your sample size calculation formula and related study.

- Have you considered medication intake for this study? And did you adjust their effects?

Results:

- Dietary intake (carbohydrates, protein, fat) as well as some micronutrients such as omega-3 fatty acids can affect blood sugar and lipid profile. Why did you not report food intake? (Please report it in table).

Discussion:

-The Discussion section is poor. Please complete your discussion using the references mentioned and related article and discuss about probably mechanism effects of this compound and also improve its grammatical errors.

Conclusion:

- Good presented.

Overall:

- I found some grammatical errors in your work. Please revise in again by a native person.

Best Regard

6. PLOS authors have the option to publish the peer review history of their article (what does this mean?). If published, this will include your full peer review and any attached files.

Reviewer #1: No

Reviewer #2: No

---

## [Author Response · Author response to Decision Letter 0]

4 Oct 2024

See attached document for our response to reviewer comments

---

## [Decision Letter · Decision Letter 1]

18 Dec 2024

PONE-D-24-00185R1Effects of Cinnamomum zeylanicum (Ceylon cinnamon) extract on lipid profile, glucose levels and its safety in adults: A randomized, double-blind, controlled trialPLOS ONE

Dear Dr. Ajanthan,

Thank you for submitting your manuscript to PLOS ONE. After careful consideration, we feel that it has merit but does not fully meet PLOS ONE’s publication criteria as it currently stands. Therefore, we invite you to submit a revised version of the manuscript that addresses the points raised during the review process.

We look forward to receiving your revised manuscript.

Kind regards,

Nishant Kumar, Ph.D

Academic Editor

PLOS ONE

Journal Requirements:

Reviewers' comments:

Reviewer's Responses to Questions

**Comments to the Author**

1. If the authors have adequately addressed your comments raised in a previous round of review and you feel that this manuscript is now acceptable for publication, you may indicate that here to bypass the “Comments to the Author” section, enter your conflict of interest statement in the “Confidential to Editor” section, and submit your "Accept" recommendation.

Reviewer #1: All comments have been addressed

Reviewer #3: (No Response)

2. Is the manuscript technically sound, and do the data support the conclusions?

Reviewer #1: Partly

Reviewer #3: Yes

3. Has the statistical analysis been performed appropriately and rigorously? 

Reviewer #1: No

Reviewer #3: I Don't Know

4. Have the authors made all data underlying the findings in their manuscript fully available?

Reviewer #1: Yes

Reviewer #3: Yes

5. Is the manuscript presented in an intelligible fashion and written in standard English?

Reviewer #1: Yes

Reviewer #3: No

6. Review Comments to the Author

Reviewer #1: The authors have put in great effort to address the comments.

Minor comment:

For Table 2, the 95% confidence interval for Cohen D is to be denoted in the table footnote.

Reviewer #3: I have carefully read through your work, and I appreciate the efforts you have put into conducting this relevant clinical trial. I have the following feedback to help strengthen your manuscript:

1. English Language and formatting consistency:

The English language used throughout the manuscript is generally clear and appropriate for a scientific publication. However, there are a few instances where the phrasing could be improved for clarity, such as the sentences in lines 378-384. Please revise carefully the whole manuscript.

- The manuscript would benefit from a thorough review of grammar, punctuation, and capitalization to ensure consistency throughout the document. For example, the term "Cinnamomum zeylanicum" should be capitalized consistently, but cinnamon shouldn’t (see, for example, line 46).

2. Abstract:

- In the objective section, you mention evaluating LDL-c levels, but you also measured other parameters, such as fasting blood sugar (FBS), that you mention on the results. It would be helpful to clarify the outcome of interest in the objective.

- Could you please provide the justification for choosing the 1000 mg/day dose of Cinnamomum zeylanicum extract? This information should also be included in the methods section (lines 149-150).

- The statement about a "significant interaction effect" (lines 22-23) could use some additional clarification.

- The two concluding sentences in the abstract should not be separated by a period.

3. Introduction:

- The reference for the sentence in line 42 is missing. Could you please provide the appropriate citation?

- It would be interesting to know the potential advantages of using Cinnamomum zeylanicum, which has a lower coumarin content, and the rationale for conducting this study.

4. Materials and methods:

- Please double-check the amount of PAC A per 100g of the cinnamon extract reported in line 142 (15298 mg of PAC A per 100g) and revise if necessary.

- For clarity, it would be helpful to define the abbreviations used in the manuscript, such as IMP (line 158), HbA1c (line 186), SBP and DBP (line 188), AST, ALT, and eGFR (line 190). Please revise all the abbreviations throughout the text.

5. Results:

- Lines 238-239: It is unclear what the "normal levels at baseline" were for the outcomes presented. Please specify the actual baseline values and what is considered the normal/healthy range for these measures.

- The presentation of Tables 2 and 3 should be improved. The number of participants per group should be clearly stated in the tables rather than using a footnote. Additionally, the formatting of the data (e.g. "Mean (SD); n (%)") should be consistent across all parameters.

- Lines 267-269: please provide the actual numerical values that support the conclusions drawn from the results, rather than just stating the observations qualitatively.

- The quality of both figures should be improved to ensure the data is clearly visible and interpretable.

6. Discussion:

- Consistency of period: Throughout the manuscript, you refer to the trial duration as both 3 months and 12 weeks. While these are equivalent, it would be best to use a single, consistent period to avoid any potential confusion.

- Strengths and limitations: I would recommend mentioning the key strengths and limitations of your study together at the end of the discussion section. This will help provide the reader with a clear overview of the merits and potential shortcomings of your work.

- Rationale for dose selection: In the discussion, you compare your results to studies using different dosages of cinnamon extract. It would be helpful to explicitly explain in the introduction or methods why you selected the 1000 mg/day dose for your trial.

- Mechanism of action: Your discussion of the putative mechanism of action for the cinnamon extract effects is interesting. If possible, can you provide any information on the specific molecules present in the extract that may be responsible for these effects?

- Liver and kidney profile details: When mentioning the liver and kidney profile findings, it would be useful to specify which parameters from Table 2 are related to these organ functions, as some readers may not be familiar with interpreting these results.

- Missing data: Please clarify what you mean by "missing data" when discussing the study limitations.

Overall, this appears to be a well-designed and conducted study that could make a relevant contribution to the literature on the metabolic effects of cinnamon supplementation. With the revisions suggested above, I believe your manuscript will be suitable for publication.

7. PLOS authors have the option to publish the peer review history of their article (what does this mean?). If published, this will include your full peer review and any attached files.

Reviewer #1: No

Reviewer #3: No

---

## [Author Response · Author response to Decision Letter 1]

5 Jan 2025

The responses to reviewer comments are enclosed as a separate document.

---

## [Decision Letter · Decision Letter 2]

7 Jan 2025

Effects of Cinnamomum zeylanicum (Ceylon cinnamon) extract on lipid profile, glucose levels and its safety in adults: A randomized, double-blind, controlled trial

PONE-D-24-00185R2

Dear Dr. Ajanthan

We’re pleased to inform you that your manuscript has been judged scientifically suitable for publication and will be formally accepted for publication once it meets all outstanding technical requirements.

Kind regards,

Nishant Kumar, Ph.D

Academic Editor

PLOS ONE

Additional Editor Comments (optional):

Reviewers' comments:

Reviewer's Responses to Questions

**Comments to the Author**

1. If the authors have adequately addressed your comments raised in a previous round of review and you feel that this manuscript is now acceptable for publication, you may indicate that here to bypass the “Comments to the Author” section, enter your conflict of interest statement in the “Confidential to Editor” section, and submit your "Accept" recommendation.

Reviewer #1: All comments have been addressed

2. Is the manuscript technically sound, and do the data support the conclusions?

Reviewer #1: Partly

3. Has the statistical analysis been performed appropriately and rigorously? 

Reviewer #1: Yes

4. Have the authors made all data underlying the findings in their manuscript fully available?

Reviewer #1: Yes

5. Is the manuscript presented in an intelligible fashion and written in standard English?

Reviewer #1: Yes

6. Review Comments to the Author

Reviewer #1: Open and close bracket missing for

Table 3 HDL-C: -1.48, 2.24

Table 4 Serum creatinine: -0.07, 0.04

7. PLOS authors have the option to publish the peer review history of their article (what does this mean?). If published, this will include your full peer review and any attached files.

Reviewer #1: No

---

## [Editor Report · Acceptance letter]

16 Jan 2025

PONE-D-24-00185R2 

PLOS ONE

Dear Dr. Ajanthan, 

I'm pleased to inform you that your manuscript has been deemed suitable for publication in PLOS ONE. Congratulations! Your manuscript is now being handed over to our production team.

Kind regards, 

on behalf of

Dr. Nishant Kumar 

Academic Editor

PLOS ONE